# Revisiting Anomaly Localization Metrics

**David Zimmerer**                    D.ZIMMERER@DKFZ.DE  and  **Klaus Maier-Hein**
K.MAIER-HEIN@DKFZ.DE
*Medical Image Computing, German Cancer Research Center (DKFZ), Heidelberg, Germany*

## Abstract

An assumption-free, disease-agnostic pathology detector and segmentor could often be seen as one of the holy grails of medical image analysis. Building on this promise, un-/weakly supervised anomaly localization approaches, which aim to model normal/healthy samples using data and then detect anything deviant from this (i.e., anything abnormal), have gained popularity. However, being an upcoming area in between image segmentation and out-of-distribution detection, most approaches have adapted their evaluation setup and metrics from either field and thus might have missed peculiarities inherent to the anomaly localization task. Here, we revisit the anomaly localization setup, discuss and analyse the properties of the often used metrics, show alternative metrics inspired from instance segmentation and compare the metrics across multiple setting and algorithms. Overall, we argue that the choice of the metric is use-case dependent, however, the Soft Instance IoU shows significant promise going forward.

**Keywords:** Anomaly Localization, Anomaly Detection, Metrics.

## 1. Introduction

Accurately detecting and localizing pathologies within medical images is a cornerstone for effective diagnosis and treatment. Unsupervised and weakly-supervised anomaly localization techniques hold potential in this arena, offering the ability to pinpoint abnormalities without extensive disease-specific labeling (Zimmerer et al., 2022b). These methods model the characteristics of normal, healthy tissue, facilitating the identification of deviations. Historically, most anomaly localization methods produce heatmaps to visualize the likelihood of anomalies within an image, necessitating specialized evaluation metrics (Schlegl et al., 2017; Baur et al., 2018; Zimmerer et al., 2018; Chen et al., 2018). However, as a rapidly developing field, anomaly localization approaches have often borrowed evaluation metrics from related domains like image segmentation and out-of-distribution (OoD) detection (Ahmed and Courville, 2019; Zimmerer et al., 2019; Marimont and Tarroni, 2020; Pinaya et al., 2021; Meissen et al., 2022; Zimmerer et al., 2022a; Lagogiannis et al., 2023). This practice might overlook the unique nuances of the anomaly localization task, potentially hindering the optimal selection of metrics.

## 2. Anomaly Localization Metrics

To effectively evaluate the performance of anomaly localization models, a range of metrics are employed, spanning various domains:

**Segmentation Metrics** Often employed metrics here are DCE (Dice Similarity Coefficient) and IoU (Intersection over Union). While commonly used in segmentation tasks (Isensee et al., 2018), DCE and IoU rely on binarized predictions. This necessitates thresholding heatmaps, a process that introduces potential bias via threshold selection and can lead to undefined scores when ground-truth segmentations are sparse – a frequent occurrence in anomaly localization

**OoD Metrics** Classical OoD Metrics are AP (Average Precision) and AUROC (Area Under the Receiver Operating Characteristic). Unlike segmentation metrics, ranking-based metrics like AUROC and AP directly handle heatmaps without requiring thresholding or relying on exact prediction values. However, they still yield undefined scores for data samples without ground-truth labels. While often addressed by combining labeled and unlabeled data (e.g., evaluating metrics across the entire dataset ("Dataset level"), or using batch-wise calculations as in Zimmerer et al. (2022a)), this approach can overemphasize larger, potentially easier-to-detect anomalies (Reinke et al. (2021); Maier-Hein et al. (2023)).

**Instance Segmentation Metrics** transition from basic overlap measurements to object-centric anomaly localization. This requires defining distinct objects in ground-truth labels (often via connected-component analysis). Key metrics include Instance IoU and Center Distance, which can be aggregated using mean, median, or by applying a threshold (e.g., IoU ¿ 0.5) to classify TPs, FPs, and FNs at the object level, enabling the calculation of derived metrics like F1-score. H owever, binarization of predictions remains necessary for object identification (e.g., using connected-component analysis). For this work, we adapt the Center Distance metric: a object's heatmap center point lying within the convex hull of a labeled object constitutes a TP.

**Anomaly Localization Metrics** To harness the strengths of instance segmentation metrics while avoiding the drawbacks of binarization thresholds, we introduce Soft Instance IoU (inspired by Soft DCE [1]). This modified Instance IoU integrates continuous anomaly scores for a more nuanced assessment of predicted anomaly confidence: $\text{SoftIoU} = \frac{\sum_{i \in Object \cup Background} \alpha \hat{y}_i y_i}{\sum_{i \in Object \cup Background}(0.5\hat{y}_i + (1-\alpha)y_i)}$, where $Background$ refers to all pixels not labeled as objects, $i$ indexes the objects in the sample and $\alpha$ is a weighting factor to balance under- and over-segmentation.

## 3. Experiments & Results

**Metric Analysis** To gain insights into metric behavior, we designed a controlled experimental setting with 50 samples using perfectly segmented, circular objects. We systematically introduced perturbations to these segmentations, including: (a) Adding small true detected objects while reducing segmentation size (roughly preserves overall segmented pixel count). (b) Varying segmentation size. (c) Adding false detections (FPs). (d) Adding missed instances (FNs). (e) Adding "empty" samples without label or prediction. A few properties of the different metrics emerge (Fig. 3, top): (a): AP and AUROC metrics unexpectedly decreased despite improved object detection. Soft IoU metrics increased as intended, while object-based metrics exhibited some noise but remained relatively consistent. (b): Most metrics exhibited the expected peak-shaped response to segmentation size

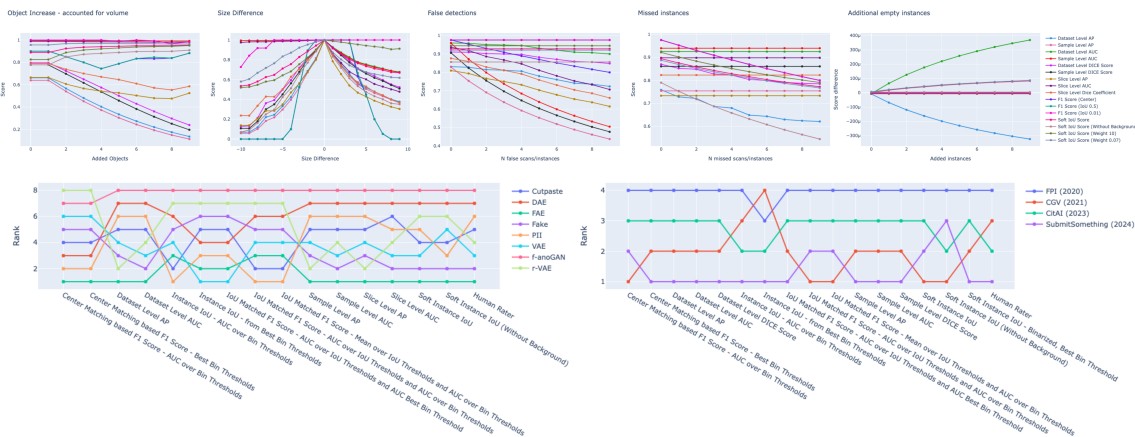

Figure 1: Top: Metrics analysis experiments. Bottom left: CamCAM, right: MOOD.

changes. However, F1 (center-based) and F1 (0.5-IoU) curves were less pronounced, while Dataset AP and F1 (0.01-IoU) were nearly constant on one side. (c): Sensitivity to false positives varied across metrics. AP, AUROC, and F1 (center) showed strong reactions, while Soft IoU was less affected. (d): F1 metrics, Soft IoU, and Dataset Level AP and DCE appeared most sensitive to missed instances. Sample-level and slice-level metrics failed to reflect the performance change. (e): Only F1 metrics and Soft IoU registered an improvement when completely normal samples were added. Dataset AP surprisingly decreased, while other metrics were insensitive by design.

**Metric Behavior in Anomaly Benchmark Settings**    In a second setting (Fig. 3, bottom), we conducted experiments to compare the performance of different anomaly detection algorithms across the diverse metrics and evaluate how well they align with human assessment in a closer to real-world setting. First, seven algorithms were tested on the CamCAM dataset (Taylor et al., 2017). We introduced artificial anomalies in the form of colored spheres (one large, four small) into 50% of the test images. The framework, hyperparameters and training schedules were kept consistent with Lagogiannis et al. (2023). Second, we evaluated the respective winning algorithms from the MOOD challenge (Zimmerer et al., 2022a) on the MOOD brain dataset, similarly introducing colored sphere anomalies in 50% of test images. Here, respectively Soft Instance IoU and F1-based metrics closely mirrored human judgment on the anomaly detection task. However, it's crucial to note that the human evaluation was restricted to segmented slices, potentially downplaying the impact of false positives unrelated to existing anomalies.

## 4. Disscussion & Conclusion

Our experiments highlight how different metrics capture distinct aspects of anomaly detection performance. While the ideal choice is task-dependent, Soft Instance IoU and F1-based metrics demonstrate favorable properties for many anomaly detection scenarios. This underscores the importance of careful metric selection to align with the goals of the specific application.

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
